# Transcriptome and chromatin landscape of iNKT cells are shaped by subset differentiation and antigen exposure

Mallory Paynich Murray [1,6], Isaac Engel[1,6], Grégory Seumois [1,6], Sara Herrera-De la Mata [1], Sandy Lucette Rosales[1], Ashu Sethi[1], Ashmitaa Logandha Ramamoorthy Premlal[1], Goo-Young Seo[1], Jason Greenbaum[1], Pandurangan Vijayanand [1,2,3], James P. Scott-Browne[1,4,7 ✉] & Mitchell Kronenberg [1,5,7 ✉]

Invariant natural killer T cells (iNKT cells) differentiate into thymic and peripheral NKT1, NKT2 and NKT17 subsets. Here we use RNA-seq and ATAC-seq analyses and show iNKT subsets are similar, regardless of tissue location. Lung iNKT cell subsets possess the most distinct location-specific features, shared with other innate lymphocytes in the lung, possibly consistent with increased activation. Following antigenic stimulation, iNKT cells undergo chromatin and transcriptional changes delineating two populations: one similar to follicular helper T cells and the other NK or effector like. Phenotypic analysis indicates these changes are observed long-term, suggesting that iNKT cells gene programs are not fixed, but they are capable of chromatin remodeling after antigen to give rise to additional subsets.

[1] La Jolla Institute for Immunology, La Jolla, CA, USA. [2] University of Southampton, Faculty of Medicine, Southampton, UK. [3] Department of Medicine, University of California San Diego, La Jolla, CA, USA. [4] Department of Biomedical Research, National Jewish Health, Denver, CO, USA. [5] Division of Biological Sciences, University of California San Diego, La Jolla, CA, USA. [6] These authors contributed equally: Mallory Paynich Murray, Isaac Engel, Grégory Seumois. [7] These authors jointly supervised this work: James P. Scott-Browne, Mitchell Kronenberg. ✉email: SCOTTBROWNEJ@njhealth.org; mitch@lji.org

Invariant natural killer T (iNKT) cells are considered to be an innate-like T lymphocyte population that can initiate or inhibit immune responses, depending on the context. Following activation, iNKT cells rapidly produce copious amounts of cytokines, similar to other innate-like lymphocytes[1]. iNKT cells express an invariant TCRα chain comprised of a Vα14-Jα18 (*Trav11-Traj18*) rearrangement in mice, with a conserved rearrangement in humans and many other mammals. These cells are activated by either self or microbial glycolipid antigens, presented by CD1d, a non-classical MHC class I molecule[2].

In the thymus, iNKT cells differentiate into three effector cell subsets, NKT1, NKT2, and NKT17, without exposure to exogenous antigen. Their effector functions and cytokine profiles resemble $T_H1$, $T_H2$, and $T_H17$ CD4$^+$ T cells and subsets of other lymphocytes, including ILC, mucosal-associated invariant T (MAIT) cells, and γδ T cells[3–5]. Within the thymus, these iNKT cell subsets have highly divergent epigenetic landscapes and transcriptional programs[6–8]. Remarkably, several hundred genes are differentially expressed between thymic iNKT cell subsets, despite their similar specificity and despite sharing a distinct positive selection pathway[9]. Evidence suggests that some iNKT cells are long-term thymic residents, and these resident cells may contribute to thymic homeostasis[6]. Following egress from the thymus, iNKT cells localize to tissues throughout the body and the majority of peripheral iNKT cells do not recirculate[10,11].

Although divergent thymic iNKT cell subsets have been identified, their relationship to the corresponding peripheral iNKT cell subsets has not been assessed. We and others have defined the molecular details of iNKT cell subset differentiation using genome-wide analyses of gene expression and chromatin accessibility[7,8,11,12]. These data identified the induction and divergence of transcriptional programs between thymic iNKT cell subsets, but the impact of tissue localization on these programs remains incompletely understood. To address these issues, we compared transcriptomic and epigenomic data of iNKT cells from the thymus to several peripheral sites. Similar methods were used to track changes in these cells after antigen exposure. Our genome-wide analysis of the transcriptome and epigenome of iNKT cell subsets provides insights into the stability and plasticity of the chromatin landscapes that are initiated in the thymus and perhaps also initiated peripherally as well.

## Results

**Divergent chromatin landscape of thymic iNKT cell subsets**. Previously, we showed that thymic iNKT cell subsets possess highly divergent transcriptomes[8]. Similar results were obtained by others[7,11,12]. Further, we demonstrated by genome-wide analysis of H3K27 acetylation modification that there were significant differences in enhancer marks between the thymic iNKT cell subsets[8]. Because the epigenetic landscape of a cell population is more stable than the transcriptome, we analyzed the epigenetic landscape of thymic iNKT cell subsets more broadly with the assay for transposase-accessible chromatin using sequencing (ATAC-seq)[13]. The thymic iNKT cell subsets were sorted based on the expression of surface proteins and validated by transcription factor staining (Supplementary Fig. 1a, b). Based on our previous RNA-seq analysis of thymic iNKT cell subsets, we excluded a population of CD1d-tetramer$^+$ cells with an intermediate phenotype, ICOS$^{high}$, or IL-17RB$^+$ cells that express CXCR3 or CD122, to obtain more purified subsets (Supplementary Fig. 1c). Expression of *Rorc* and *Tbx21* transcripts by each iNKT cell subset in multiple tissue tissues further demonstrates sorting efficiency; *Rorc* transcripts were only expressed in NKT17 cells, whereas *Tbx21* transcripts were predominately expressed in NKT1 cells (Supplementary Fig. 1d).

Consistent with previous results, we found that the profiles of accessible chromatin in iNKT cell thymic subsets were strikingly divergent, with between ~5000–7500 differentially accessible regions of chromatin (Fig. 1a). For comparison, naive versus memory CD8$^+$ T cells have ~5700 differentially accessible regions of chromatin[14,15]. Figure 1b highlights the results from some key cytokine and transcription factor gene loci. For example, there was a higher ATAC-seq signal at the *Ifng* locus in thymic NKT1 cells (Fig. 1b). Although some signal at several peaks also was apparent in NKT2 cells, no accessibility was detected at a proximal enhancer 5 kb upstream of the TSS (vertical gray bar) required for *Ifng* transcription (Fig. 1b)[16]. As expected, we found the *Il17a* locus was most accessible in NKT17 cells. The *Il4* and *Il13* loci were open in both NKT2 and NKT1 cells, likely reflecting the ability of NKT1 cells to produce some $T_H2$ cytokines after strong activation (Fig. 1b). Similarly, for transcription factors that drive the expression of key cytokines, the *Tbx21* locus encoding T-bet was more accessible in NKT1 cells and accessibility of *Rorc* was increased in NKT17 cells (Fig. 1b, right). *Zbtb16* encoding PLZF, a transcription factor required for the generation of all iNKT cells[9], was accessible in all subsets, although mRNA and protein expression (Supplementary Fig. 1b) were higher in NKT2 thymocytes. These observations from C57BL/6J thymic iNKT cells were consistent with previous analyses of gene expression and chromatin accessibility of thymic iNKT cells from BALB/c mice[7,12].

We partitioned all differentially accessible regions between thymic subsets (Fig. 1a) into eight groups with *k*-means clustering to identify potential regulatory elements with similar changes in the ATAC-seq signal. We then examined the degree to which the regions in each group were accessible in the different thymocyte iNKT cell subsets (Fig. 1c). Regions in clusters 1–3 had the highest signal in NKT1 cells, while clusters 4–5 and 6–8 had the highest signal in NKT2 and NKT17 cells, respectively. To associate the changes in regulatory element accessibility with transcription factors, we determined the enrichment of known motifs associated with DNA-binding proteins. Within clusters 1–3, typical of NKT1 cells, accessible regions frequently contained Tbox motifs, and to a lesser extent Runt and Ets motifs. HMG box protein motifs, associated with Tcf1 and Lef1, were enriched in NKT2 thymocytes along with some enrichment for zinc finger and RHD domain motifs, which include Egr1 and Nfat motifs, respectively. Expression of Tcf1 (encoded by *Tcf7*) is enriched in NKT2 cells and required for iNKT cell development[17]. Additionally, Lef1 is required for iNKT cell expansion and NKT differentiation, and NKT cell effector function, independent of Tcf1[18]. Regions accessible in NKT17 cells were enriched for consensus motifs of nuclear receptors, which can include Rorc and Reverb (Fig. 1d).

**iNKT cell subsets in different sites are similar**. To understand the degree to which the gene programs associated with thymic subsets were also present in the periphery, and to assess the impact of tissue localization on chromatin accessibility and the transcriptome, we compared sorted iNKT cell subsets from the thymus to those in the spleen, liver, and lung by both RNA-seq and ATAC-seq analysis. Previous work showed that NKT1 cells were the predominant iNKT cell population in C57BL/6J mice, comprising the majority in all the tissue sites analyzed[6]. NKT17 cells preferentially localized to the lung, lymph nodes, and skin, and NKT2 cells were more abundant in the spleen and mesenteric lymph nodes[6]. Because of the very low cell numbers, NKT2 and NKT17 cells from the liver were not analyzed. Although we used a different RNA-seq technology allowing for greater sequencing depth[19], we found exceedingly similar gene expression profiles in

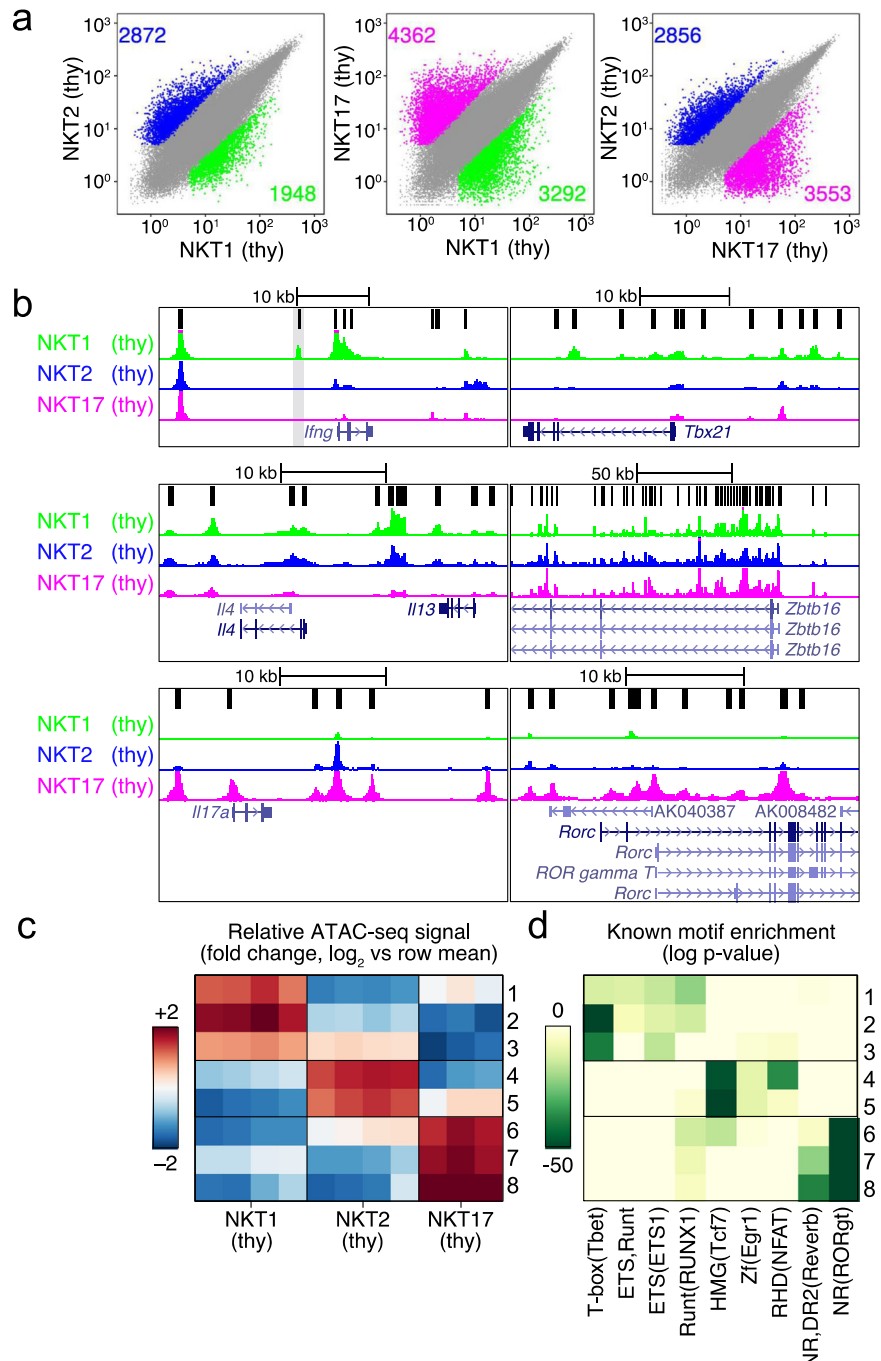

**Fig. 1 Subsets of thymic iNKT cells have differences in chromatin accessibility. a** Scatterplot of mean ATAC-seq counts per peak comparing differentially accessible regions of chromatin for pairs of thymic iNKT cell subsets. Colors indicate differentially accessible regions defined by limma/voom (details in 'Methods'). Blue = enriched in NKT2, green = enriched in NKT1, purple = enriched in NKT17. Thymus = thy. The numbers of differentially expressed genes are indicated. **b** ATAC-seq coverage at the indicated gene loci with a range of 0–600 for all samples. Gray bar in the upper left panel (*Ifng*) locus indicates the enhancer region. **c** Left, *k*-means clustering of relative ATAC-seq density (counts per million mapped reads/kb, log₂ fold change from the mean) identifies eight groups of accessible regions that varied similarly (rows), 3 sets for NKT1, 2 for NKT2, and 3 for NKT17. Columns indicate the number of replicates, 3 or 4. **d** Motifs enriched in clusters of accessible regions. All motifs with a HOMER log *p* value less than −15 using a cumulative binomial test and found in 10% or more regions in at least one cluster are shown.

thymic NKT1, NKT2, and NKT17 thymocytes compared to the previous study[8] (Supplementary Fig. 2). Based on bulk RNA-seq analysis, we observed that, as in the thymus, iNKT cell subsets within a given tissue were distinct from one another (Supplementary Fig. 2). Similarly, the chromatin accessibility profiles of iNKT subsets from the spleen showed a divergent pattern of accessible regions (Fig. 2a). Although splenic NKT2 and NKT17

cells were more similar to one another than their thymic counterparts, there remained more than 3500 differentially accessible loci (Fig. 2a).

In comparing thymic iNKT cell subsets with their peripheral counterparts, we found smaller differences related to the tissue location compared to differences that were associated with the subset identity. For example, comparing peripheral NKT1 to

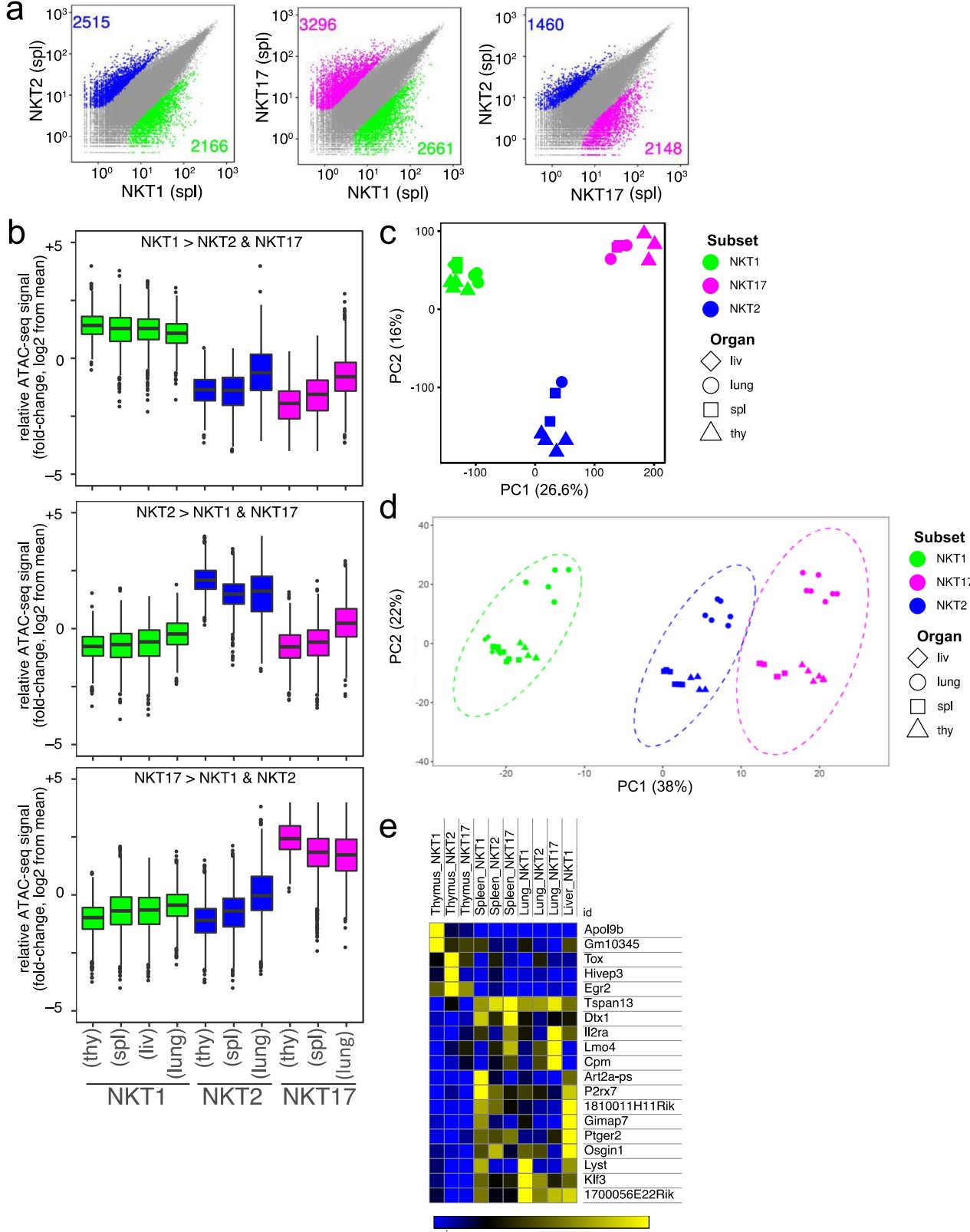

**Fig. 2 Imprint of tissue location is minor compared to the subset. a** Scatterplot of mean ATAC-seq counts per peak comparing differentially accessible regions of chromatin for pairs of iNKT cell subsets in the spleen (spl.). Colors indicate differentially accessible regions defined by limma/voom (details in 'Methods'). **b** Boxplot of normalized average ATAC-seq counts per peak from the indicated samples (labeled at bottom) at differentially-accessible regions that distinguish thymic iNKT cell subsets. Box indicates interquartile range with whiskers $+/-1.5$ times this range and outlier points indicated. $n = 4$ for thymic NKT1 and NKT2, $n = 3$ for thymic NKT17 and lung NKT1, $n = 1$ for lung NKT2, and $n = 2$ for all other samples. **c**, **d** PCA display of ATACseq (**c**) and RNA-seq (**d**) data from the indicated iNKT cell subsets from the different tissues. **e** Heat map of normalized reads from RNA-seq for genes (19) differentially regulated in thymus versus peripheral iNKT cell subsets, within all subsets, $n > 2$-fold difference, $p = 0.1$.

thymic NKT1 cells, we found less than 1000 differentially accessible chromatin regions. Similarly, we observed relatively few differences in accessible chromatin regions comparing NKT1 cells between different peripheral tissues (Fig. 2b). Similar results were obtained from analyzing chromatin accessibility in NKT2 and NKT17 cells from different tissues. Principal component analyses (PCA) of the ATAC-seq data revealed the strong influence of subset identity (Fig. 2c). A similar conclusion was obtained for the RNA-seq data, although we did find some separation based on the tissue (Fig. 2d).

Despite the overall similarity between the thymus and peripheral tissues, there are transcripts that were enriched specifically in a given thymic subset compared to the same subset in each peripheral site, including *Egr2* and *Tox* (Fig. 2e). These transcription factors are required for the early stages of iNKT cell differentiation[20,21], so-called NKT0 cells, and therefore, this may reflect the residual expression of these genes in mature thymic iNKT cell subsets, long lived in the thymus. Some transcripts were the converse, enriched in all or several peripheral tissues compared to the thymus, without subset restriction. These include *Art2* and *P2rx7*, previously reported to be increased in total populations of peripheral iNKT cells[22], which make cells sensitive to NAD-induced cell death and *Osgin1*, identified as a growth inhibitory protein in other contexts[23]. The expression of these genes might reflect the need for brakes on the expansion and function of potentially autoreactive iNKT cells[24]. Also enriched in all sites in peripheral iNKT cells were *Tspan13*, and *Klf3*, whose expression was increased in memory CD8$^+$ T cells[25].

**Identification of a gene expression signature in lung**. Although the iNKT cell subset is a predominant factor driving genomic differences, lung iNKT cells shared common features that distinguished them from their counterparts in the other locations. This was revealed by PC2 of the RNA-seq data (Fig. 2d), or PC3 analysis of the ATAC-seq data (Fig. 3a). Transcripts encoding AP-1 and other bZIP family members, as well as some members of the NF-κB family were enriched in all lung iNKT cell subsets, as were transcripts encoding CTLA-4, CD69, and *Nr4a1* encoding Nur77 (Fig. 3b and Supplementary Fig. 3). Furthermore, regions more accessible in lung iNKT cells were enriched for bZIP motifs, which are associated with the transcription factors Ap1 and Atf, as well as RHD motifs, which can include NF-κB-p65-binding sites (Fig. 3c). Together these data are consistent with tissue-residency or increased activation of lung iNKT cells, and indeed, we found a subpopulation of lung iNKT cells that expressed CTLA-4 by flow cytometry (Fig. 3d). To ascertain that the signature in lung iNKT cells could not be attributed to infection or inflammation in a single mouse, we sorted iNKT cell subsets from individual mice obtained only one week earlier from a commercial supplier and performed RNA-seq. We found a similar lung gene expression signature in each individual (Supplementary Fig. 3).

Cells within the lung are exposed to a diverse environment of environmental and microbial antigens, as well as differences in oxygen. We next asked if iNKT cells from another antigen-rich site, the small intestine, displayed a similar increase in CTLA-4 expression. We found that total iNKT cells from the small intestinal lamina propria (SI-LPL) expressed CTLA-4 similarly to lung iNKT cells, whereas splenic iNKT cells did not (Fig. 3e). These data suggest antigen-rich environments may imprint aspects of what we are calling the lung activation signature in different sites.

Other innate or innate-like lymphocyte populations are found in the lung, including γδ T cells, MAIT cells, ILC, and NK cells, as well as mainstream resident lymphocytes and some circulating

cells. We tested if the lung activation signature of iNKT cells extended to several other lung populations. Therefore, we performed ATAC-seq and RNA-seq analyses on sorted γδ T cells, NK cells, as well as naive CD4$^+$ T cells from the lung and spleen. Lung γδ T cells and NK cells displayed the lung signature based on increased chromatin accessibility in regions enriched for bZIP and RHD motifs (Fig. 4a), with γδ T cells having accessibility in regions enriched for nuclear receptor motifs, which can include RORγt-binding sites, while NK cells were enriched for T-box motifs. Lung CD4$^+$ T cells had a different pattern from the other cell types, but with some increased signal at regions containing bZIP (ATF) and RHD (NF-κB-p65) motifs (Fig. 4a). PCA analysis of the RNA-seq data, which included total iNKT cells from the spleen or lung, showed separation of each lung cell type, including CD4$^+$ T cells, from the corresponding cell type in the spleen (Fig. 4b). Further, we found increased expression of the iNKT cell lung signature genes, listed in Supplementary Fig. 3, in lung γδ T cells, and NK cells compared to the corresponding splenic populations (Fig. 4c). The transcriptome of lung CD4$^+$ T cells was more divergent, but still had some features in common with the lung-resident innate or innate-like lymphocyte populations (Fig. 4c). This is illustrated in Fig. 4d, which shows that expression of *Fosl2*, *Bhlhe40*, and *Tnfaip3* was higher in all cell types from the lung. Gene set enrichment analysis (GSEA)[26] pre-ranked analysis comparing each cell type from the lung versus spleen using the iNKT cell lung signature further demonstrated the strong enrichment of the lung signature in each cell type (Fig. 4e).

**Epigenomic and transcriptomic changes following antigen**. A unifying hypothesis based on these data is that iNKT cell subsets are formed in the thymus and seed peripheral tissues with fixed functions and relatively minor impacts to their chromatin landscape and transcriptomic profiles, with the partial exception of those in the lung. To test this, we determined if these profiles remained after a strong antigenic challenge in total splenic iNKT cells, which are mostly NKT1 cells. To examine how the chromatin landscape and transcriptome of iNKT cells were altered in response to antigen, we injected mice with αGalCer and harvested the spleen 6 days later. It has been reported that following exposure to the potent glycolipid antigen α-galactosylceramide (αGalCer), some iNKT cells display a T follicular helper cell (T$_{FH}$)-like phenotype, with increased expression of CXCR5, PD-1, and BCL6 (Supplementary Fig. 4a, b). These so-called NKT$_{FH}$ cells produce IL-21, and localize to germinal centers[27], and may play a role in early germinal center formation[28], but their gene expression programs had not been elucidated. Putative NKT$_{FH}$ (αGalCer-loaded CD1d tetramer$^+$CXCR5$^+$PD-1$^+$) and the remaining population of iNKT cells from antigen-injected mice (CXCR5$^-$PD-1$^-$ or NKT non-FH) cells were sorted and analyzed by ATAC-seq and RNA-seq. Of note, antigen-experienced versions of the NKT1, NKT2, and NKT17 cells as described above were not identifiable 6 days following αGalCer challenge and therefore could not be analyzed separately. As shown in Fig. 5a, accessible regions of chromatin were exceptionally different comparing NKT$_{FH}$ and the antigen-exposed non-FH cells, which we refer to as iNKT cell effectors (NKT$_{eff}$). Chromatin accessibility regions in NKT$_{FH}$ were also very different from the NKT1, NKT2, and NKT17 subsets in the spleen from uninjected mice, most different from NKT17 cells (Fig. 5a, left). There was increased chromatin accessibility in NKT$_{FH}$ in the *Il21* locus and the *Pdcd1* locus encoding PD-1 in NKT$_{FH}$ cells (Fig. 5b), reflecting a T$_{FH}$ state. The accessible regions of chromatin within NKT$_{eff}$ also greatly varied when compared to the iNKT cell subsets from unchallenged mice but were most similar to NKT1 cells

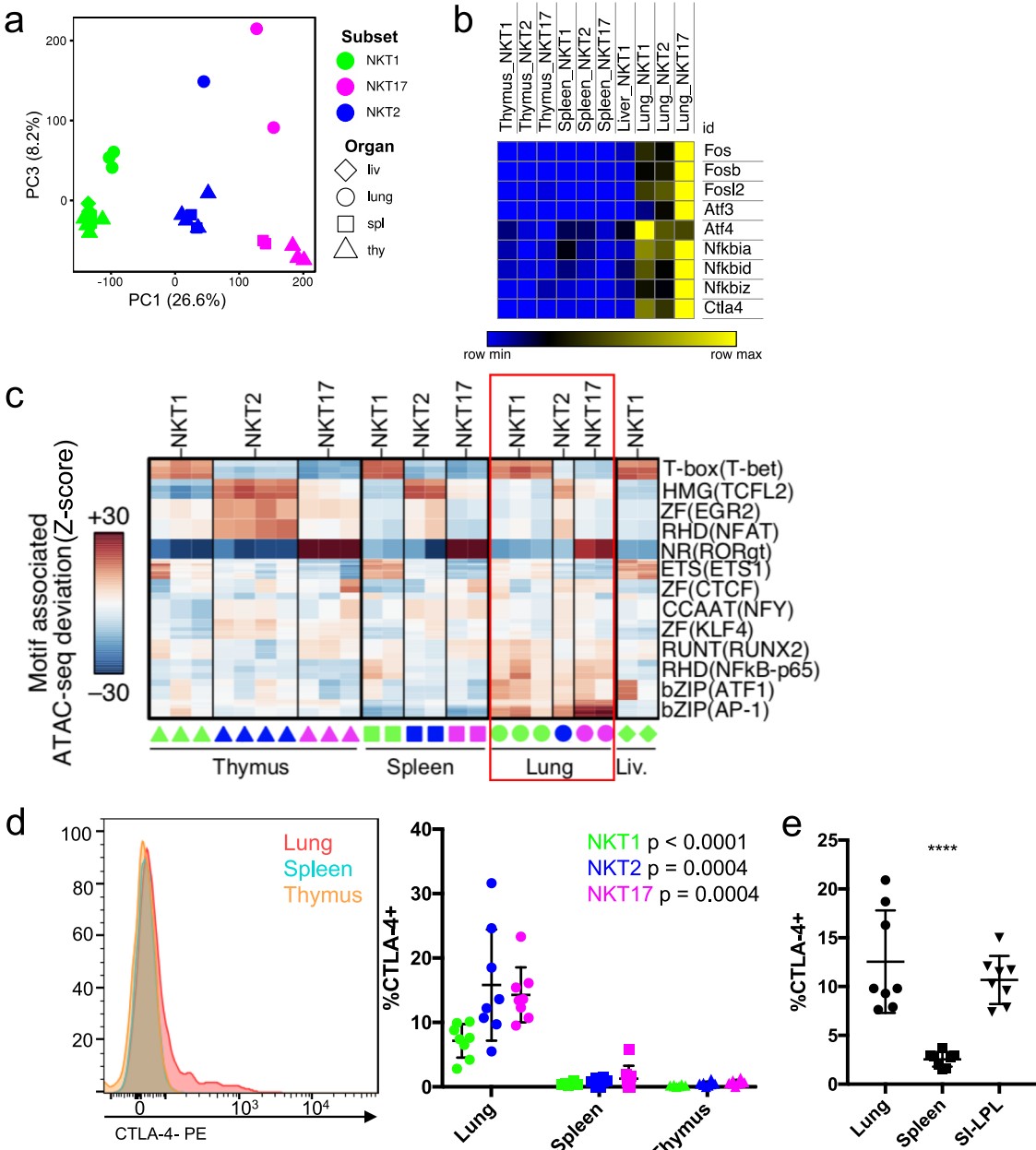

**Fig. 3 Lung-specific transcriptome and epigenome. a** PC1 by PC3 display of ATAC-seq data showing the distribution of iNKT cell subsets from different tissues. Liver = liv. **b** Heat map of relative RNA expression of selected AP-1 and ATF family genes in iNKT subsets from the indicated sites. **c** chromVAR computed deviation in ATAC-seq signal (Z-score) at regions containing indicated transcription factor motifs. Motifs with a $p$ value less than 1e-25 are shown and families and representative members are labeled. Samples are indicated at the bottom, iNKT cell subsets from the lung are boxed to highlight lung-enriched motifs. **d** Total CTLA-4 expression in permeabilized iNKT cells from the indicated tissues. Representative cytogram of total iNKT cells (left) and percent CTLA-4+ cells within each subset from the different organs (right). Symbols depict individual mice, bars depict mean and SD. Data are combined from five experiments, $n = 8$ mice, statistical significance ($p < 0.001$ for NKT1 and $p = 0.0004$ for NKT2 and NKT17) assessed via Kruskal–Wallis tests. **e** CTLA-4 expression in permeabilized total iNKT cells from indicated tissues. Symbols depict individual mice, bars depict mean and SD, $n = 8$ mice from two independent experiments. Statistical significance ($p < 0.0001$) assessed via one-way ANOVA.

(Fig. 5a, right). These data suggest that most of the splenic iNKT cells were exposed to the antigen, including those that did not become NKT_FH.

We partitioned all differentially accessible regions between NKT_FH and NKT_eff cells and splenic iNKT cell subsets from unimmunized mice into ten groups with $k$-means clustering to identify regions with similar changes in ATAC-seq signal. As above, we examined the degree to which the regions in each group were differentially accessible in iNKT cell populations (Fig. 5c) and their association with DNA-binding protein motifs (Fig. 5c). Notably, the two populations from αGalCer immunized mice had increased accessibility for cluster 7, with motifs for T-box proteins and to a lesser extent NF-κB (RHD motifs) and IRF proteins. There also was decreased accessibility in regions containing motifs associated with the lineage driving transcription factors RORγt (nuclear receptor motifs) and GATA3 (zinc finger motifs) enriched in clusters 1 and 2. Accessible regions specific to NKT_FH cells within clusters 8 and 9 were increased for motifs for RHD domain transcription factors, which include NFAT and NF-κB, and bZIP motifs,

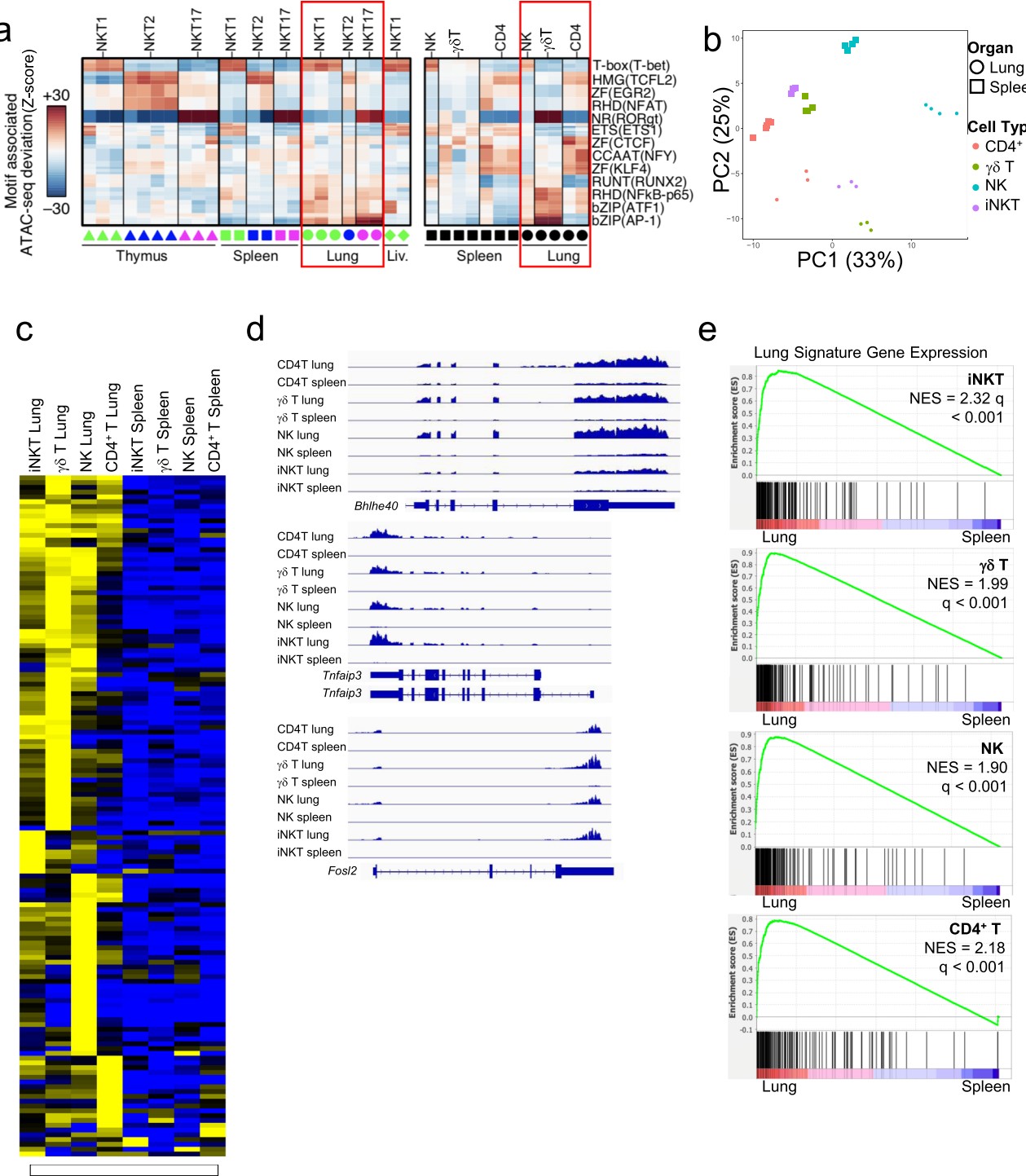

**Fig. 4 Lung signature extends to other lung populations. a** chromVAR computed deviation in ATAC-seq signal (Z-score) at regions containing indicated transcription factor motifs between the indicated cell populations from spleen and lung. Motifs with a *p* value less than 1e-25 are shown and families and representative members are labeled. Tissue source of the samples indicated at the bottom of the diagram; columns indicate number of replicates. **b** PCA of RNA-seq data comparing lymphoid cell types isolated from lung and spleen. **c** Heat map depicting the lung gene signature from Supplementary Fig. 3 with normalized transcript levels from RNA-seq data from the indicated cell types from lung and spleen. **d** RNAseq read tracks in the *Bhlhe40*, *Tnfaip3*, and *Fosl2* loci in samples from lung and spleen from indicated cell types. **e** GSEA of pre-ranked comparisons of genes differentially expressed in lung iNKT cell subsets comparing each cell type within the lung to its counterpart in the spleen.

characteristic of Ap-1 family transcription factors, suggesting a more activated state. Clusters 4 and 5 were more accessible in $NKT_{eff}$ and also in NKT1 cells from unimmunized mice. These accessible regions were enriched for T-bet, Ets, and Runt domain-associated motifs. $NKT_{eff}$ cells also had increased

accessibility in regions containing more Zinc finger transcription factor and Ets motifs (cluster 6). Although they are reported to be self reactive, these data suggest that iNKT cells greatly remodel their chromatin landscape following an encounter with a potent exogenous glycolipid antigen.

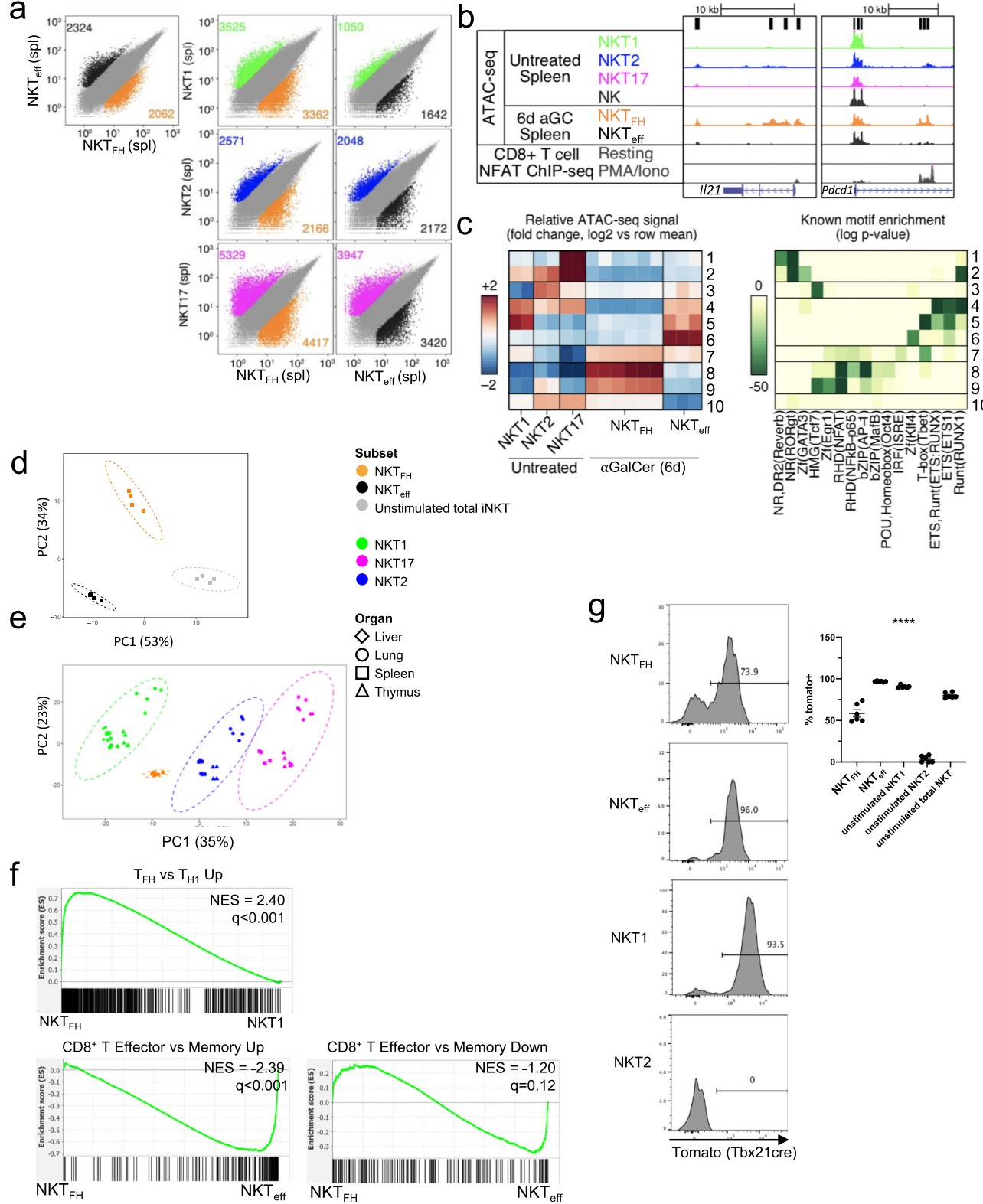

We also performed RNA-seq on the two populations of iNKT cells from αGalCer-treated mice, as well as total splenic iNKT cells from unimmunized mice. Gene expression by NKT$_{FH}$ was greatly different from total splenic iNKT cells (Fig. 5d) or from each of the iNKT cell subsets from unchallenged mice (Fig. 5e and Supplementary Fig. 4c). Comparison of differentially expressed genes distinguishing CD4$^+$ T$_{FH}$ and T$_H$1 cells[29] demonstrated that NKT$_{FH}$ shared a more similar expression profile to T$_{FH}$, whereas, not surprisingly, NKT1 cells were similar to T$_H$1 (Supplementary Fig. 4d). Similarities between NKT$_{FH}$ and T$_{FH}$ also were revealed by GSEA analysis (Fig. 5f, top panel). GSEA of NKT$_{FH}$ compared to NKT$_{eff}$ revealed an enrichment for CD8$^+$ T cell effector-related genes in NKT$_{eff}$ (Fig. 5f, bottom, left, and right panels). Neither antigen-experienced subset was significantly enriched for memory cell-related signatures.

**Fig. 5 Antigenic experience shapes the transcriptome and chromatin landscape of iNKT cells.** iNKT cells from αGalCer-injected mice either selected for PD-1 and CXCR5 expression indicating similarity to follicular helper (FH) T cells (NKT$_{FH}$) or negative for both markers and similar to effectors (eff) (NKT$_{eff}$) were sorted from the spleen of mice 6 days after injection (i.v.) with αGalCer. **a** Scatterplots of mean ATAC-seq counts per peak comparing antigen-experienced NKT$_{FH}$ and NKT$_{eff}$ (top left) or pairwise comparisons of differentially accessible regions of chromatin for each of the sorted populations from antigen exposed mice compared to the corresponding subsets from unimmunized mice. Data from unimmunized mice are the same as depicted in Fig. 2. Colors indicate differentially accessible regions defined by limma/voom (details in Methods). **b** ATAC-seq coverage (range of 0–600 for all samples) comparing the *Il21* and *Pdcd1* loci from unimmunized splenic iNKT cell subsets, splenic NK cells, and NKT$_{FH}$ and NKT$_{eff}$ from αGalCer-treated mice. NFAT ChIP-seq analysis of CD8$^+$ splenic T cells with and without PMA/ionomycin stimulation included for comparison[62]. **c** Left, *k*-means clustering of relative ATAC-seq density (counts per million mapped reads/kb, log$_2$ fold change from the mean) identifies ten groups of accessible regions that vary similarly (rows), two sets for splenic NKT1, two for splenic NKT2, two for splenic NKT17, six for NKT$_{FH}$, and three for NKT$_{eff}$. Columns indicate the number of replicates. Right, motifs enriched in clusters of accessible regions. All motifs with a HOMER log *p* value less than –15 and found in 10% or more regions in at least one cluster are shown. **d** PCA analysis of RNA-seq data comparing NKT$_{FH}$ to NKT$_{eff}$ from αGalCer-immunized mice, as well as to total iNKT cells from unimmunized mice. **e** PCA analyses of RNA-seq data comparing splenic NKT1, NKT2, or NKT17 samples to spleen NKT$_{FH}$ cells. **f** Top: Plot of the distribution of genes upregulated in mainstream GC T$_{FH}$ vs T$_H$1 in a list of genes ranked by relative expression (directional *p* value) in NKT$_{FH}$ vs splenic NKT1 cells using GSEA. Bottom (left and right): Plots of genes differentially regulated between CD8$^+$ effector vs memory against a directional p-ranked file comparing αGalCer-stimulated NKT$_{FH}$ vs NKT$_{eff}$. Normalized Enrichment Scores (NES) and *q* values were determined by the pre-ranked GSEA algorithm. **g** Expression of reporter in T-bet fate-mapping mice by NKT$_{FH}$ and NKT$_{eff}$ cells 6 days post antigen exposure, and NKT1 and NKT2 cells and total iNKT cells from unstimulated mice; *n* = 6 mice per group, error bars depict SEM. Quantification on right, statistical significance (*p* < 0.0001) assessed via Kruskal–Wallis test.

Because of the great prevalence of NKT1 cells in the spleens of unimmunized C57BL/6J mice[6], it was not feasible to assess directly the separate contributions of the NKT1, NKT2, and NKT17 subsets to the antigen-activated iNKT cell populations. To address their origin, we utilized mice in which T-bet expression could be fate mapped[30]. Whereas close to 90% of spleen NKT1 cells from unimmunized mice had expressed T-bet according to the fate mapping mice, only 74% or less of NKT$_{FH}$ did at day 6 (Fig. 5g). Notably, T-bet expression was absent in NKT2 cells. Decreased marks of previous T-bet expression in NKT$_{FH}$ were observed as early as day 3 after antigen (Supplementary Fig. 4e). Because prior T-bet expression was selected against in NKT$_{FH}$ cells, these data suggest that NKT2 and/or NKT17 cells contributed to the NKT$_{FH}$ pool. Furthermore, the fate mapping may underestimate how efficiently NKT2 and/or NKT17 converted to NKT$_{FH}$; T-bet expression could have been induced in some iNKT cells after antigen activation, although we have no evidence that this occurred. Unless a minority population expanded greatly and converted rapidly and transiently to T-bet expression, some NKT$_{FH}$ likely also originated from the prevalent NKT1 cell pool. Also, we note that despite their separation from NKT1 cells, we found expression of some NKT1 signature genes by NKT$_{FH}$ (Supplementary Fig. 4d).

**Enhanced effector function after antigen challenge**. As shown in Fig. 5a (right column), iNKT cells from αGalCer-immunized mice that did not become NKT$_{FH}$ also had a chromatin landscape different from all of the subsets in unimmunized mice, with the biggest divergence again from NKT17 cells. PCA analysis of the RNA-seq data showed that the transcriptome of these iNKT cells was highly different from total iNKT cells from unimmunized mice (Fig. 5d). Previous analyses have shown that after i.v. exposure to DCs loaded with αGalCer, a KLRG1$^+$ population of iNKT cells develops, especially in the lung, in a process dependent on expression of the transcription factor Eomes[31,32]. Cells with this phenotype persisted for weeks and they exhibited enhanced effector function. Pathway analysis of genes enriched in NKT$_{eff}$ in the spleen following αGalCer alone, using the ConsensusPath Database[33], identified NK cell-mediated cytotoxicity as the most enriched pathway (Supplementary Fig. 4f). This is in line with the gene expression profile in these lymphocytes indicative of an enhanced effector phenotype found by GSEA (Fig. 5f). Intriguingly, there was a similarity in chromatin accessibility in some key loci between NKT$_{eff}$ and splenic NK cells, with increased ATAC-

seq peaks within the loci for genes encoding Granzyme A and B, KLRG1, and CX3CR1, as well as Spry2, which is also highly expressed by NK cells (Fig. 6a). There also were some regions of increased chromatin accessibility in genes associated with NK cell function in NKT1 cells, but these regions had higher signals in NKT$_{eff}$ and NK cells.

To validate the existence of the NKT$_{eff}$ population, we assessed the expression of KLRG1 and CX3CR1, NK cell markers with increased chromatin accessibility in NKT$_{eff}$ (Fig. 6a). We detected increased expression of each of these markers on NKT$_{eff}$ compared to NKT$_{FH}$ and iNKT cells from uninjected mice (Fig. 6b). T-bet fate mapping analysis revealed that virtually all KLRG1$^+$ NKT$_{eff}$ had expressed T-bet, suggesting either these cells differentiated from NKT1 cells or acquired expression T-bet when activated (Fig. 5g). To determine if the phenotypic changes we observed were maintained, we also analyzed splenic iNKT cell populations at day 30 or later after antigen exposure. iNKT cells with the NKT$_{eff}$ phenotype were still a sizeable fraction of the iNKT cells (Fig. 6c). Similarly, NKT$_{FH}$ cells also persisted in the spleen beyond day 30 (Supplementary Fig. 4g), consistent with a report showing elevation of NKT$_{FH}$ 60 days post treatment with αGalCer and ovalbumin-loaded liposomes[34]. These data suggest that antigen challenge induces dynamic and prolonged changes in the phenotype of iNKT cells reflecting changes in the transcriptome and chromatin landscape.

## Discussion

There are functional subsets of iNKT cells, analogous to CD4 T$_H$1, T$_H$2, and T$_H$17 cells, as well as several other lymphocyte populations[7], and it has been established that the chromatin landscape and transcriptomes of the thymic iNKT cell subsets are distinct[7,8,12]. Here, we addressed three questions. First, to what extent are the gene programs driving the thymic iNKT cell subsets present in peripheral iNKT cells? Second, considering that iNKT cells are mostly non-recirculating lymphocytes[11], what is the imprint of localization in different tissues on these gene programs? Third, to what extent are the iNKT cell gene programs subject to dynamic and long-term changes following antigenic stimulation, as such changes might be suggestive of trained immunity or an effector-memory response?

Our data indicated that the status of chromatin accessibility and the transcriptome in any one subset are relatively similar to one another, regardless of location. Although this might suggest that iNKT cells become fully mature and committed to a subset in

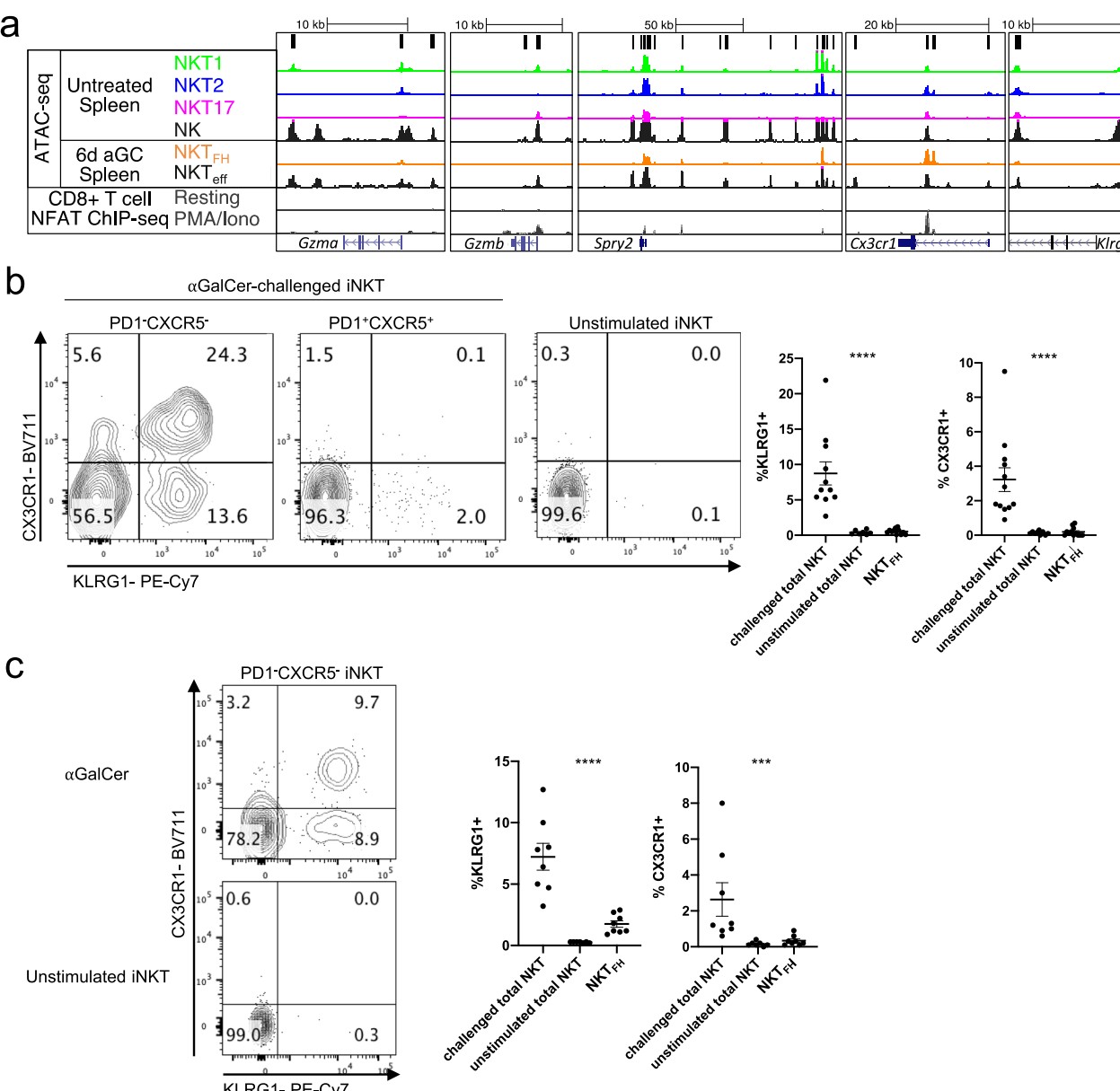

**Fig. 6 Enhanced effector or NK cell signature in antigen-exposed iNKT cells. a** ATAC-seq coverage (with a range of 0–600 for all samples) at the *Gzma*, *Gzmb*, and *Spry2* loci comparing untreated splenic iNKT cells subsets, spleen NK cells, NKT$_{FH}$, and NKT$_{eff}$ from mice injected 6 days earlier with αGalCer. NFAT1 ChIP-seq analysis of CD8$^+$ T cells with and without PMA/ionomycin stimulation included for comparison[62]. **b** Flow cytometry analysis of expression of KLRG1 and CX3CR1 by the indicated populations of gated spleen iNKT cells at day 6 after antigen. Quantification on right, $n = 11$ mice (αGalCer challenged, KLRG1), $n = 12$ mice (αGalCer challenged, CXCR3), $n = 6$ mice (unstimulated, KLRG1), $n = 8$ mice (unstimulated, CXCR3), $n = 11$–12 mice (NKT$_{FH}$). **c** (Left) Representative flow cytometry analysis of expression of NKT$_{eff}$ markers CX3CR1 and KLRG1 by spleen PD-1$^-$ KLRG1$^-$ iNKT cells at day 30 or later after antigen administration. (Right) Quantification of iNKT cells expressing NKT$_{eff}$ markers, challenged mice analyzed at day 30 or later. $n = 8$ mice (αGalCer challenged, NKT$_{FH}$), $n = 7$ mice (unstimulated). **b**, **c** Representative data from 3–5 experiments, error bars depict mean and SEM, statistical significance (***=$p$ of 0.005, ****=$p < 0.0001$) assessed via Kruskal–Wallis test.

the thymus and maintain their status in the periphery, there is evidence based on the expression of diagnostic surface proteins that iNKT cell recent thymic emigrants are not fully mature[35,36,37]. To align these facts, perhaps the epigenome of the iNKT cell subsets is set up in the thymus prior to emigration, but the mature, subset-specific transcriptomes only initiated after thymus emigration. ATAC-seq analysis of stage 0, stage 1, and mature NKT1 cells identified regions accessible in mature NKT1 cells that were already accessible in stage 0 cells, providing some evidence to support this hypothesis[38]. Alternatively, it is possible that the recent thymic emigrants are not fully

differentiated even with regard to their chromatin landscapes into functional subsets, but instead, receive tissue-specific cues allowing the cells to mature into effector subsets resembling their thymic counterparts.

Although the imprint of tissue localization was comparatively limited, in the lung iNKT cells exhibited motif enrichment in regions of accessible chromatin for bZIP domain transcription factors, which can include AP-1 and ATF, and RHD (NF-κB) transcription factors, regardless of the functional subset. This may be related to an activation signature, consistent with the increased CTLA-4 expression by lung iNKT cells. These data are consistent

with a recent report describing a similar gene expression signature in lung MAIT and iNKT cells, although it was also deemed to be consistent with a tissue-residency pattern[11]. Regardless, the lung signature was present not only in iNKT cell subsets, but also in NK cells and γδ T cells, and to a lesser extent even in CD4[+] T cells compared to the corresponding cell type in the spleen. A recent study comparing the epigenome of alveolar CD8[+] resident memory T cells (Trm) found that Trm cells within the lung interstitial space were enriched for AP-1, FOS, and CREM motifs compared to splenic Trm[39]. In other studies, this lung signature was not just specific to lymphocytes, with some similarities to the lung epigenome of alveolar macrophages[40]. Based on these findings and the present study, the lung microenvironment may dictate epigenetic remodeling and subsequent transcriptional changes. One potentially important factor is increased oxygen concentration[41,42], and also, the lung may face more environmental exposure to external substances and microbes[42]. Perhaps immune cells in the lung need to be poised to rapidly respond to challenges. If this were correct, then we would predict that iNKT cells in sites such as skin or intestine might also have gene programs distinct from those in thymus, spleen, and liver. Consistent with this hypothesis, we found increased CTLA-4 expression by iNKT cells from the SI-LPL. Additionally, it has been found that iNKT cells from the draining lymph nodes of the skin and the small intestine, the inguinal lymph node, and mesenteric lymph node, expressed a number of lung signature genes, including *Fos*, *Fosb*, and *Nr4a1*, compared to those in thymus and spleen[37]. A more detailed exploration of the so-called lung activation signature in iNKT cells and other innate populations within different antigen exposed tissues is needed.

Six days after antigenic exposure, we detected two different iNKT cell populations, one is NKT$_{FH}$ that is similar to T$_{FH}$. Previously, NKT$_{FH}$ were reported, based on the expression of BCL-6, a few key surface proteins, and functional assays for T cell help[27]. Here, we demonstrated that this NKT$_{FH}$ population has a dramatically different chromatin landscape and transcriptome that resembles T$_{FH}$. These cells likely originated in part from NKT1, the major splenic subset population, but probably also from other activated subsets, considering the evidence for reduced prior T-bet expression. Furthermore, they persisted for at least 30 days after antigenic challenge. At later time points, NKT$_{FH}$ maintains the capacity to produce IL-21, but down-regulated BCL-6 expression and increased expression of CD62L and CCR7, consistent with memory T$_{FH}$[34]. As detected by immune assays, the half-life of αGalCer complexes with CD1d on the surface of DCs in vivo was less than 24 h[43,44]. Given this half-life, it is unlikely that these cells experience any recent exposure to αGalCer. Therefore, chromatin remodeling in iNKT cells after antigen exposure led to the generation of a persisting population of iNKT cells that is expected to have enhanced helper function for B cells.

The second population called NKT$_{eff}$ more closely resembled NK cells. iNKT cells with this phenotype also were present for at least 30 days. Previously a similar population was present in the lung after injection of antigen-loaded DCs[32], but here we show that NKT$_{eff}$ can be generated systemically following the same antigenic challenge that also induces the NKT$_{FH}$ population. In other experiments, it was reported that iNKT cells exposed to αGalCer were anergic for many weeks[45,46]. In previous work, we did not find evidence for anergy in the total iNKT cell population, because, at 30 days following αGalCer, re-challenged iNKT cells remained cytotoxic, effectively signaled through their TCR, and had increased proliferation compared to iNKT cells responding to αGalCer for the first time[47]. However, we did find reduced pro-inflammatory cytokine production, which could reflect the distinct functions of NKT$_{FH}$ cells that were likely generated, and a

minority of the iNKT cells that acquired the ability to produce IL-10[47]. Therefore, it is likely that iNKT cells from mice injected with the antigen that did not become NKT$_{FH}$ are heterogenous, including not only NKT$_{eff}$ but IL-10 producers and perhaps some anergic cells. Further, in studies of KLRG1[+] iNKT cell induction by antigen-loaded DCs, these cells maintain long-term anti-tumor function[32]. The size of the NKT$_{eff}$ population is difficult to quantify because KLRG1[+] cells accounted for only a portion, with other cells expressing high amounts of CX3CR1 and granzymes (Fig. 6b, d). The contributions of TCR signal strength, co-stimulation, cytokines, and other aspects to the generation of disparate iNKT cell populations remain to be determined, but apparently, pulsing DC with αGalCer is a method for generating NKT$_{eff}$ almost exclusively in the lung[32].

iNKT cells have been classified as innate-like T lymphocytes that bridge the innate and adaptive immune systems, sharing components of cells within each major branch of the immune response[5]. The capacity for innate-like cells to exhibit long-term changes in their functional programs in response to antigen exposure has been referred to as trained immunity[48]. Do iNKT cells exhibit an effector memory response or a form of trained immunity? At steady-state, iNKT cells have some properties of tissue-resident memory cells. Many of them express molecules characteristic of canonical resident memory T cells, such as CD103 and CD69, and like effector memory cells, they produce cytokines in a TCR-dependent or cytokine-dependent manner within a few hours[9,11]. In response to αGalCer, however, dynamic, long-term changes occur in iNKT cells, generating NKT$_{FH}$, NKT$_{eff}$ populations, and likely others, indicating a degree of plasticity and heterogeneity that allow these cells to adapt to their history of prior antigenic stimulation and respond in a variety of contexts.

## Methods

**Mice**. C57BL/6J female mice aged 6–8 weeks old were purchased from Jackson Laboratories or bred in-house at La Jolla Institute for Immunology. B6;CBA-Tg (Tbx21-cre)1Dlc/J (Tbet-cre) were purchased from Jackson Laboratories and then bred with B6.Cg-Gt(ROSA)26Sor$^{tm14(CAG-tdTomato)Hze}$/J (Td-tomato) mice (also obtained from Jackson Laboratories) to generate the T-bet fate mapping line. All studies were approved by the Institutional Animal Care and Use Committee at the La Jolla Institute for Immunology.

**Tissue preparation**. Following euthanasia, thymus tissue and spleens were removed, lungs and livers were perfused with 3–10 mL of liver perfusion medium (Gibco) until tissues cleared. Livers were mashed through a 70 μM nylon filter (Fisher). Liver lymphocytes were isolated by centrifugation at $850 \times g$ in 37.5% Percoll for 20 min. Lungs were placed in GentleMacs C tubes (Miltenyi Biotec) with 2 mL Spleen Dissociation Medium (STEMCELL Technologies) and homogenized using the Miltenyi GentleMacs dissociator. Following homogenization, suspensions were filtered with a 70 μM filter and washed twice with RPMI + 10% FBS. Thymus and spleens were homogenized through a 70 μM nylon filter and washed with RPMI + 10% FBS. Small intestinal lamina propria lymphocytes (SI-LPL) were isolated as previously described[49]. Briefly, small intestines were collected from mice and Peyer's patches were removed. The tissue was washed and cut into pieces that were then incubated in 25 ml of HBSS containing 25 mM HEPES and 5 mM EDTA in a shaker at 225 r.p.m., 37 °C, 3–5 times for 10 min. Then, tissues were incubated at 37 °C for 25 min with rotation in media containing collagenase type VIII (Sigma). The filtered cell suspension was re-suspended in 40% Percoll solution and overlaid above 80% Percoll solution. LPL were collected from the interface.

**Antigen challenge**. α-galactosylceramide (αGalCer or KRN7000) was supplied as a lyophilizate in a vehicle and provided by Kyowa Kirin Pharmaceutical Research. It was resuspended at 200 μg/mL in water and then diluted to 10 μg/mL in PBS. 0.2 mL of 10 μg/mL αGalCer was administered via retro-orbital injection, and 6- or ≥30-days later mice were euthanized and organs harvested.

**Cell sorting**. For RNA-seq and ATAC-seq experiments, unless otherwise noted iNKT cell subsets were sorted in parallel from tissues pooled from 15–20 female C57BL/6J mice, ∼6 weeks of age. Cell suspensions from thymus, spleen, and lung were enriched for iNKT cells by negative selection using biotinylated antibodies against CD8α (53–6.7, BD Biosciences), CD19 (1D3, Tonbo Biosciences), CD24

(M1/69, BD Biosciences), CD62L (MEL-14, Invitrogen), CD11b (M1/70, Tonbo Biosciences), CD11c (N418, Tonbo Biosciences), F4/80 (BM5.1, Tonbo Biosciences), EpCam(G8.8, BioLegend), and TER-119 (TER-119, Tonbo Biosciences) together with Rapidspheres (StemCell technologies) and either the Big Easy or Easy eight magnets (StemCell technologies) using protocols from the manufacturer. The remaining cells were then suspended at $10^8$ cells/mL, and incubated with 1 μg/mL of Streptavidin A (Sigma-Aldrich). Liver lymphocytes were not enriched for iNKT cells. iNKT cells were stained using a 12-parameter panel of reagents including tetramers of CD1d loaded with αGalCer (BV421, in house preparation), live/dead yellow (ThermoFisher Scientific), anti-TCRβ-APC-eF780 (H57-597, ThermoFisher Scientific), anti-CD8α-PE CF594 (53–6.7, BD Biosciences) and anti-CD19-PE CF594 (1D3, BD Biosciences), anti-CD4-AF700 (GK1.5, BioLegend), anti-IL-17RB-AF488 (FAB10402G, R&D Systems), anti-ICOS-PerCP Cy5.5 (C398.4 A, BioLegend), anti-CD122-BV650 (5H4, BD Biosciences), anti-CXCR3-APC (CXCR3-173, BioLegend), anti-SDC1-PE (281-2, BioLegend), and anti-FR4-PE Cy7 (ebio12A5, ThermoFisher Scientific). Cells were sorted using a FACSAria III or FACSAria Fusion (BD Biosciences) for live lymphocytes, singlets, CD8−CD19−, Tetramer+TCRβ+ iNKT cells and separated into NKT1, NKT2, and NKT17 cell subsets based on the following expression profiles: NKT1: CXCR3+ICOS−CD122+SDC1−IL-17RB−; NKT2: CXCR3−ICOS+CD122−SDC1−IL-17RB+CD4+; NKT17: CXCR3−ICOS+CD122−SDC1+IL-17RB+FR4−.

For NKT$_{FH}$ sorts, spleens were harvested 6 days following the antigen challenge. Single-cell suspensions were enriched for iNKT cells as described above. iNKT cells were stained with CD1d tetramers loaded with αGalCer, live/dead yellow, anti-CD8α-PE CF594 and anti-CD19-PE CF594, anti-TCRβ-APC-eF780. NKT$_{FH}$ cells were identified based on the expression of CXCR5 (anti-CXCR5−PE, clone L138D7, BioLegend), and PD-1 (anti-PD-1-APC, clone RMP1-30, BioLegend). CXCR5−PD-1− iNKT cells from antigen challenged mice were also sorted for comparison (NKT$_{eff}$).

For sorting of γδ T cells, CD4+ T cells, NK cells, and iNKT cells from lung and spleen, tissues were prepared and enriched as described above. Populations were sorted based on the following gating strategy: live lymphocytes, singlets, CD8−CD19−; iNKT cells, Tetramer+TCRβ+; γδ T cells, TCRβ−TCRγδ+ (anti-TCRγδ-FITC, clone GL3, BD Biosciences); NK cells, TCRβ−, TCRγδ−, NK1.1+ (anti-NK1.1-PE Cy7, clone PK136, BD Biosciences); CD4+ T cells, TCRβ+CD4+.

**Flow cytometry.** Cells isolated from the lung or spleen were stained for iNKT cell subsets, NKT$_{FH}$, and NKT$_{eff}$, as described above. In addition to the antibodies and other reagents used for iNKT isolation described above, we also used anti-PLZF-AF647 (clone R17-809), anti-T-bet-AF488 (clone O4-46), anti-RORγt-PE-CF594 (clone Q31-378), anti-BCL-6-AF488 (clone K112-91), all from BD Biosciences, anti-KLRG1- PECy7 (clone 2F1), and anti-CTLA-4-PE (clone UC10-4B9), all from Thermo-Fisher Scientific, and anti-CX3CR1-BV711 (clone SA011F11, BioLegend). Staining for intracellular antigens, including PLZF, T-bet, RORγt and total CTLA-4 was performed by incubating in 50% Cytofix (BD Biosciences) in PBS for 15–30 minutes on ice followed by washing and staining in eBioscience permeabilization buffer (Thermo-Fisher Scientific)[50]. Stained samples were analyzed using a Fortessa flow cytometer (BD Biosciences) and FlowJo software (Treestar).

**RNA-seq.** Cells representing specific iNKT subsets were sorted by pools of cells ranging between 200 to 400 directly into 0.2 ml PCR tubes containing 8 μl of low-input lysis buffer (0.2% Triton-X-100 and RNase inhibitor) and stored at −80 ºC until processed further. For thymic subsets, $n = 5$ (NKT1), $n = 4$ (NKT2), $n = 6$ (NKT17). For peripheral subsets, NKT1: lung $n = 5$, liver $n = 5$, spleen $n = 7$; NKT2: lung $n = 6$, spleen $n = 7$; NKT17: lung $n = 7$, spleen $n = 6$. For NKT$_{FH}$, $n = 6$, NKT$_{eff}$, $n = 3$. For lung cell types, NK: lung $n = 3$, spleen $n = 4$; γδ T cells: lung $n = 3$, spleen $n = 3$; CD4+ T cells: lung $n = 3$, spleen $n = 5$. For bulk library preparation for sequencing, we used the Smart-Seq2 protocol, adapted for samples with small cell numbers[51]. We followed the protocol with following modifications[19]: (i) the pre-amplification PCR cycle was set between 17 to 23 cycles; (ii) to eliminate any traces of primer-dimers, the PCR pre-amplified cDNA product was purified using 0.8X Ampure-XP beads (Beckman Coulter) before using the DNA for sequencing library preparation[19]. One ng of pre-amplified cDNA was used to generate barcoded Illumina sequencing libraries (Nextera XT library preparation kit—Illumina) in 8 μl reaction volume. Samples failing any quality control step (DNA quality assessed by capillary electrophoresis (Fragment analyzer, Advance analytical) and quantity (Picogreen quantification assay, Thermofisher) were eliminated from further downstream steps[19]. Libraries were then pooled at equal molar concentration and quantified (KAPA SYBR® FAST qPCR Kit—Roche)[19]. Samples from the experiment comparing splenic NKT$_{FH}$ and NKT$_{eff}$ cells from αGalCer-challenged mice with unchallenged splenic iNKT cells were sequenced via a 100 × 100 bp paired-end read strategy using the NovaSeq6000 sequencing platform (NovaSeq 6000 S4 P200 kits—Illumina). Sequencing for all other samples was performed according to a 50 bp single-end strategy using the HiSeq2500 sequencer (HiSeq SBS Kit v4; Illumina). Post-sequencing, stringent quality controls were applied and samples that failed quality control standards were eliminated from further analysis[19]. Samples we sequenced to obtain at least 8 million uniquely mapped reads.

The single-end reads that passed Illumina filters were filtered for reads aligning to tRNA, rRNA, adapter sequences, and spike-in controls. The reads were then aligned to mm10 reference genome using TopHat (v 1.4.1)[52]. DUST scores were calculated with PRINSEQ Lite (v 0.20.3)[53] and low-complexity reads (DUST > 4) were removed from the BAM files. The alignment results were parsed via the SAMtools[54] to generate SAM files. Read counts to each gene were obtained with the htseq-count program (v 0.7.1)[55] using the "union" option. After removing absent features (zero counts in all samples), the raw counts were then imported in most cases to the R/Bioconductor package DESeq2 (v 1.6.3)[56] to identify differentially expressed genes among samples, with $P$ values for differential expression calculated using the Wald test for differences between the base means of two conditions. For the analysis of iNKT samples collected after in vivo stimulation, the R/Bioconductor package EdgeR was used[57], and $P$ values were determined using the quasi-likelihood F test. For determination of signature genes of specific iNKT subsets, we included genes that exhibited a twofold or greater expression difference with a $P$ value of ≤0.1 in all pairwise comparisons between the given subset and the other two subsets in each of the organs from which all three subsets were collected (thymus, spleen, and lung). Similarly, the signature genes of spleen, lung, and thymus were defined as those genes exhibiting a fold change of ≥2 with a $P$ value of ≤0.1 in all possible comparisons between the same iNKT subset from different organs. Signature genes of splenic iNKT populations that either expressed or lacked the FH markers PD-1 and CXCR5 after αGalCer stimulation were determined in a similar fashion, except that $P$ values were adjusted for multiple test correction using Benjamini–Hochberg algorithm[58], and both stimulated populations were also compared to unchallenged total splenic iNKT. Principal Component Analysis (PCA) was performed using the 'prcomp' function in R. Data were also analyzed using the Pre-ranked Gene Set Enrichment Analysis algorithm (Broad Institute and University of California), as well as the Consensus Path Database platform (Max Planck Institute).

**ATAC-seq.** ATAC-seq was performed as previously described with some modifications[13]. For thymic subsets, $n = 4$ (NKT1), $n = 4$ (NKT2), $n = 3$ (NKT17). For peripheral subsets, NKT1: lung $n = 3$, liver $n = 2$, spleen $n = 2$; NKT2: lung $n = 1$, spleen $n = 2$; NKT17: lung $n = 2$, spleen $n = 2$. For NKT$_{FH}$, $n = 6$, NKT$_{eff}$, $n = 3$. For lung cell types, NK: lung $n = 1$, spleen $n = 2$; γδ T cells: lung $n = 2$, spleen $n = 3$; CD4+ T cells: lung $n = 2$, spleen $n = 3$. iNKT cells or lung lymphocytes (10,000) were sorted into 1.5 mL Eppendorf tubes containing PBS with 5% FCS. Cells were centrifuged at 600 $g$ for 10 min at 4 ºC, washed with 50 μL PBS, then resuspended in 50 μL ATAC lysis buffer (10 mM Tris pH 7.5, 10 mM NaCl, 3 mM MgCl$_2$, 0.1% NP-40). Cells were centrifuged in lysis buffer for 10 min at 600 $g$, 4 ºC. Following lysis, the pellet was resuspended in 50 μL ATAC reaction mix (25 uL 2X TD buffer, 2.5 μL Nextera Enzyme, 22.5 μL water, Illumina). The transposase reaction was carried out at 37 ºC for 30 min. Libraries were amplified using a KAPA HiFi real-time library amplification kit with barcoded primers for 11–12 cycles followed by 2 × 50 cycle paired-end sequencing. Reads were mapped to the mouse genome (mm9) using bowtie. Unmapped reads were processed with trim galore, re-mapped with bowtie, and merged with previous mapping output. Duplicate reads identified by picard MarkDuplicates and reads mapping to chrM were excluded. Wiggle files of coverage for individual replicates were computed with MEDIPS[59] using full fragments captured by ATAC-seq on 10 bp windows and used to generate average coverage with the Java Genomics Toolkit (available at: https://github.com/timpalpant/java-genomics-toolkit) for each group. Accessible regions were identified using MACS2[60] from individual replicate bam files downsampled to a maximum of 5 million reads and limited to a $q$ value of less than 0.001. Peaks that intersected ENCODE blacklisted regions and those on chromosome Y were excluded. We refined the groups of accessible regions to non-overlapping peaks with a uniform width of 500 nucleotides with the readNarrowpeaks function from chromVAR[61]. The number of reads within each region was computed using all reads from each replicate with the getCounts function from chromVAR. Differentially accessible regions were identified with limma/voom, using quantile normalized counts, and selected based on an fdr adjusted $p$ value of less than 0.1 and an estimated fold change of at least four. We associated transcription factors-binding motifs from the HOMER database by determining the enrichment of motifs in groups of peaks with HOMER and comparing the variability in ATAC-seq signal with chromVAR.

**Reporting summary.** Further information on research design is available in the Nature Research Reporting Summary linked to this article.

## Data availability statement

Sequence data that support the findings of this study are deposited in the Gene Expression Omnibus (accession code GSE161492). ATAC-seq sequence data associated figures: Fig. 1; Fig. 2a–c; Fig. 3a, c; Fig. 4a; Fig. 5a–c; Fig. 6b. RNA-seq sequence data associated figures: Fig. 2d; Fig. 3b; Fig. 4b–e; Fig. 5d–f; Fig. 6a; Supplementary Fig. 2; Supplementary Fig. 3; Supplementary Fig. 4c, d. There are no restrictions on data availability. Additional information and materials will be made available upon request.

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

## Acknowledgements

We thank Shu Liang, Alice Wang, Monalisa Mondal, and Jeremy Day for assistance with next generation sequencing and the Flow Cytometry Core Facility at the La Jolla Institute for Immunology for assistance with cell sorting. We thank Archana Khurana for generating CD1d-αGalCer tetramers. We thank Anjana Rao for critical feedback on the manuscript. Supported by the US National Institutes of Health R01 AI71922, AI105215, AI137230 to M.K.; R01 AI040127 and AI109842 to Anjana Rao and Patrick Hogan; T32 AI125179 to M.P.M.; Shared Instrumentation Grant (SIG) Program S10 OD018499 to the Flow Cytometry Core Facility at the La Jolla Institute for Immunology; S10 RR027366 for a FACSAria II cell sorter to Dr. Michael Croft; S10 OD016262 for an Illumina HiSeq 2500 sequencing system to Dr. Anjana Rao; S10 OD025052 for a NovaSeq 6000 sequencer to Dr. Gregory Seumois.

## Author contributions

M.P.M., I.E., and G.S. designed and performed experiments and analyzed data. S.H.M., S.L.R., and G.-Y.S. performed experiments. A.S., A.L.R.P., and J.G. analyzed the RNA-seq data. P.V. designed the study. J.S.B. analyzed the ATAC-seq data. M.P.M. and M.K. wrote the manuscript. J.S.B. and M.K. supervised the project, designed the study, and analyzed data.

## Competing interests

The authors declare no competing interests.
