## [Peer Review File · Nature Communications]

REVIEWER COMMENTS

Reviewer #1 (Remarks to the Author):

This manuscript by Murray et al. characterizes the accessibility of chromatin in NKT cell subsets in multiple tissues under homeostatic, activation, and post-activation conditions using ATAC-sequencing. They also correlate these accessibility regions with the transcriptome in the different NKT cell subsets. They conclude that under homeostatic conditions, NKT cell subsets (NKT1, NKT2 and NKT17) have very divergent epigenomes but that each subset is similar in different tissues (i.e. NKT1 in different tissues are similar). The most divergence within a lineage is seen in the lungs and the divergent gene program is enriched in other innate lymphocyte populations indicating that this is something programmed by the lung environment. By grouping ATAC-seq patterns across the 3 subsets they identify subset specific patterns and bioinformatically identify enriched transcription factor motifs. These data suggest that T-box, Ets, and Runx binding sites are prevalent in NKT1 gene loci, HMG/TCF1, Zf and RDH binding sites are prevalent in NKT2 genes and NR (nuclear hormone receptor) sites are prevalent in NKT17 genes. They show that aGalCer stimulation induces an NKTfh signature, as noted previously by others, but that these NKTfh cells arise from both T-bet expressing (NKT1) and non-expressing (other NKT subsets). The activated cells then take on an enhanced effector program associated with increased expression/accessibility at cytotoxic effector genes such as *Gzma*, *Gzmb*, and express *CX3CR1*, *KLRG1* that persists for up to 30 days. These data indicate that activation permanently impacts the epigenome of NKT1 cells.

Overall, this is a very interesting manuscript with high quality data that provides substantial insight into NKT cell biology. It is well-written and easy to follow with an excellent methods section. I have only a few minor comments regarding presentation, references and some of the discussion points.

1. The authors note that NKT2 are enriched for TCF-1 binding sites and that TCF-1 is required for NKT2 development. This is an interesting point because there is evidence that LEF-1 is also expressed in, and required for development of NKT2, independent of TCF-1. (Carr et al., JEM 212:793, 2015). The authors should mention this point rather than just focusing on TCF-1 and reference this paper. (ex. Line 118-121)
2. What is the earliest time point that the authors looked at to assess Tbx21 reporter expression in NKTfh cells? Is it possible that all NKTfh cells arise from Tbx21 negative progenitors and then some differentiate into Tbx21+ cells? Maybe if you look early you can see this? I do not think the authors need to do this experiment but if they have the data it would be of interest. Similarly, do fully formed NKT1 cells have the ability to generate NKTfh?
3. In the discussion the authors do a nice job of discussing the possible origins of tissue NKT cell subsets. They seem to prefer the hypothesis that differentiated cells leave the thymus i.e. that liver NKT1 arise from thymic NKT1. I infer this from their comment at the end of the introduction starting on Line 69 "Our genome-wide analysis of the transcriptome and epigenome of iNKT cell subsets provides insights into the stability and plasticity of gene expression programs that are initiated in the thymus". I am not sure if they mean to say this because in the discussion they talk about the possible precursors that leave the thymus and may differentiate in tissues, but still generating cells with similar chromatin accessibility profiles. On line 343 they say "Perhaps the epigenome of the iNKT cell subsets is set up in the thymus prior to emigration, but with the mature, subset-specific transcriptome only initiated after thymus emigration". This hypothesis would be consistent with data from Wang and Hogquist, *Elife*, 7 (2018) and a very recent paper in *Nature Immunology* seems to suggest that this epigenetic program could be set up in ST0 in a BCL-6 dependent process (Gioulbasani et al, *Nat. Imm.* Jul 27, 2020 ahead of print). It might be useful to point out these papers in this context to support the early epigenetic priming of future NKT lineages.

4. Figure 6A is too small to read and could be moved to the supplemental data. The labels on graphs in this figure as also much too small

Reviewer #2 (Remarks to the Author):

In the current manuscript Paynich Murray et al. analyzed the chromatin landscape for the various iNKT cell subsets found in different tissues. Using RNAseq and ATAC-seq the authors show that the landscapes are distinct for NKT1, NKT2 and NKT17 cells. Regarding the tissue of origin, only lung NKT cells possessed location-specific features that were not detected for cells found in spleen or liver. Antigen stimulation altered the chromatin and transcriptional profiles of NKT cells, leading to the appearance of NKT cells with features of TFH or NK cells. This data suggests that NKT cells are capable of altering their chromatin and transcriptional profiles in response to antigen triggering.

Overall this is a very interesting study that provides novel insights into the regulation and pliability of NKT cell populations. Experiments are correctly executed, figures are clear and it reads well. I have some questions/comments explained below:

1) One key question relates to the gating strategy for NKT cell sorting. Surface marker expression is known to be different for NKT cells found in different tissues, so the authors should show how the gating/transcription factor expression compares for various tissues

2) The authors propose that "the chromatin accessibility patterns and transcriptomic profiles set up for iNKT cells in the thymus largely carry over into the periphery". However, recent studies show the presence of NKT cell precursors, which are found in various tissues and terminally differentiate in the periphery (Wang et al; Jimeno et al. eLife). It is not clear how both models fit. While the landscape of NKT1s may be similar in spleen/thymus/liver/lung there seem to be still differences associated with the tissue of origin, which are particularly obvious in the lung, but seem to be present also in NKT cells from the other tissues (PCoA in Fig 2C-2D). I am not convinced that the authors can exclude that precursors differentiate in the periphery on the basis of their results.

Also, can the authors identify NKT cell precursors with their gating? I wonder whether precursors are being included with the NKT2 population (as they are enriched within PLZFhi cells).

3) The effect of the tissue environment on the NKT cell transcriptome/chromatin landscape is more obvious in the lung (in comparison with spleen and liver). To understand the effect of the environment on NKT cells, it would be more relevant to check NKT cells from other organs such as the intestine or lymph nodes, in which cells would be exposed to very unique and "antigen rich" environments and which are particularly enriched in NKT2/NKT17 (absent from liver and found at low numbers in spleen)

4) The data related to changes in NKT cells after antigen exposure is interesting. It is unclear why the authors decide to compare "NKTFH" cells and "NKT effector cells" and they are not looking at NKT1, NKT2, NKT17 before/after activation.

While the populations of "NKTFH" cells and "NKT effector cells" are phenotypically different, whether they are functionally different is not shown. Also, "NKTFH" cells are enriched in the spleen 30 days after antigen administration. Do these cells maintain an increased capacity to function as TFH cells?

Mallory Paynich Murray et al., “Transcriptome and Chromatin Landscape of iNKT cells are Shaped by Subset Differentiation and Antigen Exposure”

Response to Reviewers

We would like to thank the reviewers for their appreciation of the value of our work and for their very thoughtful critique. Here we have included our point-by-point reply:

Reviewer 1.

1. The authors note that NKT2 are enriched for TCF-1 binding sites and that TCF-1 is required for NKT2 development. This is an interesting point because there is evidence that LEF-1 is also expressed in, and required for development of NKT2, independent of TCF-1. (Carr et al., JEM 212:793, 2015). The authors should mention this point rather than just focusing on TCF-1 and reference this paper. (ex. Line 118-121)

Sorry to have omitted this paper--we have now cited it (lines 127-8).

2a. What is the earliest time point that the authors looked at to assess Tbx21 reporter expression in NKT_{FH} cells? Maybe if you look early you can see this? I do not think the authors need to do this experiment but if they have the data it would be of interest.

The reviewer brings up several excellent points that we have broken down into three parts. First, we have found that NKT_{FH} and NKT_{eff} cells appear as early as 3 after α GalCer. Based on Tbx21 fate mapping experiments, we find similar percentages of Tbx21 fate map positive cells at day 3 as we did at day 6 (compare new Supplementary Fig. 4E to Figure 5G in main manuscript, see lines 297-299).

2b. Is it possible that all NKT_{FH} cells arise from Tbx21 negative progenitors and then some differentiate into Tbx21+ cells?

Second, we cannot formally rule this possibility out beyond any doubt, but we consider it unlikely for several reasons. In mainstream CD4⁺ T cells T-bet is a negative regulator of T_{FH} formation. For examples see *J Exp Med.* (2011) 208:1001–13 and *Nat Immunol.* (2012) 13:405–11. Of course, NKT cells could be different. However, considering the large number of NKT_{FH} that are generated by day 3, and that approximately 75% are T-bet fate map positive, the minor starting

Fig. 1. T-bet staining of spleen T cells.
 1 = unstimulated NKT cells.
 2 = CD4 T cells.
 3-6 are day 6 α GalCer stimulated NKT cells.
 3 = NKTFH (light green plot)

1. Unstimulated NKT1
2. CXCR3⁺CD122⁺ CD4⁺ T
3. CXCR5⁺ PD-1⁺ iNKT NKT_{FH}
4. KLRG1⁻ CX3CR1⁻ CXCR5⁻ PD-1⁻ iNKT
5. KLRG1⁺ CX3CR1⁻ CXCR5⁻ PD-1⁻ iNKT
6. KLRG1⁺ CX3CR1⁺ CXCR5⁻ PD-1⁻ iNKT

population of fate-map negative “naïve” NKT cells would have to undergo a vast expansion and T-bet conversion within three days. This conversion would have to be transient as well, because NKTFH have lower levels of T-bet protein than other antigen-exposed or activated NKT cells

(Fig.1 for reviewers). Therefore, we consider such a wholesale conversion unlikely.

2c. Similarly, do fully formed NKT1 cells have the ability to generate NKT_{fh}?

Third, as above, considering the rapid generation of NKT_{fh} and also that the majority are fate map positive, it is very likely that many of the NKT_{fh} precursors were fully formed NKT1. In other words, the most likely explanation is that fate map positive NKT_{fh} cells were positive before antigen exposure, and those that were negative remained so. Perhaps like CD4⁺ T_{fh}, different functional NKT cell subsets can become follicular helpers—but we agree that we have not ruled out other interpretations. Regarding 2b and 2c, these points are now discussed in lines 303-307.

3. In the discussion the authors do a nice job of discussing the possible origins of tissue NKT cell subsets. They seem to prefer the hypothesis that differentiated cells leave the thymus i.e. that liver NKT1 arise from thymic NKT1. I infer this from their comment at the end of the introduction starting on Line 69 “Our genome-wide analysis of the transcriptome and epigenome of iNKT cell subsets provides insights into the stability and plasticity of gene expression programs that are initiated in the thymus”. I am not sure if they mean to say this because in the discussion they talk about the possible precursors that leave the thymus and may differentiate in tissues, but still generating cells with similar chromatin accessibility profiles. On line 343 they say “Perhaps the epigenome of the iNKT cell subsets is set up in the thymus prior to emigration, but with the mature, subset-specific transcriptome only initiated after thymus emigration”. This hypothesis would be consistent with data from Wang and Hogquist, *Elife*, 7 (2018) and a very recent paper in *Nature Immunology* seems to suggest that this epigenetic program could be set up in ST0 in a BCL-6 dependent process (Gioulbasani et al, *Nat. Imm.* Jul 27, 2020 ahead of print). It might be useful to point out these papers in this context to support the early epigenetic priming of future NKT lineages.

We have clarified the writing to make it clear that the preponderance of the evidence indicates that recent thymic emigrants are not differentiated into functional subsets by phenotype and we have cited the papers mentioned by the reviewer (lines 71-72; 362-369). We like the hypothesis that the epigenome is set up in the thymus, and there are some data in favor of this (Gioulbasani et al—new reference 36), but this will not be conclusive until NKT cell recent thymic emigrants have been characterized at the level of their epigenome.

4. Figure 6A is too small to read

We have fixed this problem—thanks for pointing it out. We moved Figure 6A to Supplemental Figure 4E where we believe it fits better.

Reviewer 2.

1. One key question relates to the gating strategy for NKT cell sorting. Surface marker expression is known to be different for NKT cells found in different tissues, so the authors should show how the gating/transcription factor expression compares for various tissues

We now have included the gating strategy for the different tissues in Supplementary Figure 1A. Staining for transcription factor expression (Supplementary Figure 1B) and RNA-Seq analyses for transcription factor mRNA (Supplementary Figure 1D) confirm the purification of the NKT cell subsets from the different tissues.

2. The authors propose that “the chromatin accessibility patterns and transcriptomic profiles set up for iNKT cells in the thymus largely carry over into the periphery”. However, recent studies show the presence of NKT cell precursors, which are found in various tissues and terminally differentiate in the periphery (Wang et al; Jimeno et al. eLife). It is not clear how both models fit. While the landscape of NKT1s may be similar in spleen/thymus/liver/lung there seem to be still differences associated with the tissue of origin, which are particularly obvious in the lung, but seem to be present also in NKT cells from the other tissues (PCoA in Fig 2C-2D). I am not convinced that the authors can exclude that precursors differentiate in the periphery on the basis of their results. Also, can the authors identify NKT cell precursors with their gating? I wonder whether precursors are being included with the NKT2 population (as they are enriched within PLZFhi cells).

We agree with the critic and have clarified the writing to make it clear that the preponderance of the evidence indicates that recent thymic emigrants are not differentiated into functional subsets and have included Jimeno et al (new reference 36). Therefore, it is in fact not likely that transcriptomes are set up in the thymus and maintained during thymus egress and homing (see lines 71-2, 362-269). We like the hypothesis that the epigenome is set up in the thymus and is present in recent thymic emigrants before the transcriptome is set up, but this is only a hypothesis, and it is possible that the signals received and transcriptomes induced for NKT cell subset commitment in the periphery and thymus are similar.

Fig. 2 GSEA analysis of some known NKT0 genes comparing splenic NKT1 to NKT2.

In the thymus, it can be difficult to separate NKT2 cells from precursors. We compared the genes with increased expression in thymic NKT0 cells from single cell analysis (Engel et al., *Nat Immunol* 17: 728, 2016) to spleen NKT1 and NKT2 cells. In Fig. 2 for the reviewers and editors, we show GSEA analysis comparing these genes increased in NKT0 cells to splenic NKT1 vs. NKT2. We do find a slight bias towards enrichment for NKT0 genes in NKT2 cells, although it was not statistically significant. Therefore, the contribution of iNKT cell progenitors to our analysis of peripheral NKT2 cells likely is minimal.

3. *The effect of the tissue environment on the NKT cell transcriptome/chromatin landscape is more obvious in the lung (in comparison with spleen and liver). To understand the effect of the environment on NKT cells, it would be more relevant to check NKT cells from other organs such as the intestine or lymph nodes, in which cells would be exposed to very unique and “antigen rich” environments and which are particularly enriched in NKT2/NKT17 (absent from liver and found at low numbers in spleen)*

Thanks for this suggestion. The possibility of other antigen-rich locations also inducing the “lung activation signature” in NKT cells is intriguing. Therefore, we tested if the lung signature gene CTLA-4 was increased in iNKT cells from small intestinal lamina propria by flow cytometry. We found increased expression of CTLA-4, similar to the lung. These data are presented in new Figure 3E and discussed in the manuscript (lines 197-203 and 392-399). Further detailed analyses similar to those we have done will be required to examine NKT cells in different sites for common antigen exposure or tissue resident gene programs. However, indeed these new data suggest there could be common programs induced in NKT cells in response to the environment.

4. *It is unclear why the authors decide to compare “NKTFH” cells and “NKT effector cells” and they are not looking at NKT1, NKT2, NKT17 before/after activation. While the populations of “NKTFH” cells and “NKT effector cells” are phenotypically different, whether they are functionally different is not shown. Also, “NKTFH” cells are enriched in the spleen 30 days after antigen administration. Do these cells maintain an increased capacity to function as TFH cells?*

Treatment of mice with α GalCer induces dynamic changes in iNKT cells and we did not find recognizable, activated NKT1, NKT2, nor NKT17 subsets following challenge (see lines 245-247). In principal, adoptive transfers experiments of each iNKT cell subset could have been performed to identify the extent to which different subsets have the capacity to differentiate into NKT_{FH} and NKT_{eff}. Unfortunately, in our hands, these experiments have proved technically difficult, especially considering the low frequency of NKT2 and NKT17 cells in B6 mice and the relatively low number of NKT cells that can be recovered following transfer even of total NKT cells. Furthermore, most of the transferred cells localize to the liver, where we find much fewer NKT_{FH}. Therefore, we used the fate-mapping mice which provided insight by showing that fate map positive cells were significantly decreased in NKT_{FH} compared to NKT_{eff}.

In regard to the functionality of NKT cells with the NKT_{FH} and NKT_{eff} phenotypes at 30 days post-treatment, previous studies have shown that NKT_{FH} at later time points maintain the capacity to produce IL-21 and help B cells (*J Immunol*, 200, 3117-3127, 2018), whereas NKT_{eff} provide anti-tumor immunity (*Proc Natl Acad Sci U S A*, 111, 12474-12479, 2014). These studies are referenced in the manuscript.

We are grateful for the opportunity to modify the manuscript and now we have made a complete and thorough response to the reviewers' comments. We look forward to your reply and hope to hear back from you soon.

Sincerely yours,

Mitchell Kronenberg, Ph.D.

REVIEWERS' COMMENTS:

Reviewer #2 (Remarks to the Author):

The authors have addressed all my comments and concerns and made appropriate changes in the manuscript and the figures. Overall, this is an elegant and well-executed work. I think that this manuscript is suitable for publication in Nat Comms.